# Stocking Density and Homogeneity, Considerations on Pandemic Potential

Max J. Moreno-Madriñan [1,*] and Eric Kontowicz [2]

[1] Global Health Program, DePauw University, 602 S College Ave, Greencastle, IN 46135, USA
[2] Department of Comparative Pathobiology, College of Veterinary Medicine, Purdue University, West Lafayette, IN 47907, USA
* Correspondence: mjmorenom@depauw.edu; Tel.: +1-765-658-5090

**Simple Summary:** There is a growing trend of outbreaks of zoonotic diseases worldwide. Many of the diseases jumped to humans from domestic animals. A combination of many factors can explain this process. Industrial farming is a concern because of its characteristically high population density of susceptible hosts of single species. Such characteristic seems favorable to the transmission and evolution of pathogens. It may also resemble our increasing human population density in urban centers. This article describes how high population density of single species may integrate with other factors to increase the risk of mutation, reassortment, and the generation of new pathogens. And how current urban conditions may resemble that factor common to factory farms. Understanding these processes is essential to avoid their consequences.

**Abstract:** Zoonotic pathogens, an increasing threat to human health, typically originate in the wild but spill over to humans from domestic animals because of the high contact with them. Industrial farming involves an increased number of animals of a single species per given area. Such high stocking density facilitates pathogen transmission. This speeds evolution and also offsets the natural tendency of pathogens to trend toward mildness. On the other hand, close contact reduces transmission dependence on host mobility and thus could favor virulence. Forestalling this problem requires understanding opportunities for spillovers and evolution created by animal farming technologies and human-animal-ecosystem interactions. This manuscript considers two important risk factors of intensive animal farming, stoking density and homogeneity, to inform practices that could stop the next pandemic at its source.

**Keywords:** pathogens; evolution; agriculture; farm; urbanism; population density





## 1. Introduction

While global mortality and morbidity linked to infectious diseases declined in the past two decades thanks to advances in sanitation, medicine, and public health, the number of infectious disease outbreaks is increasing, the most recent being the COVID-19 pandemic [1]. These occurring outbreaks are caused by pathogens that increasingly spill over to humans from mammals [2], including, likely, SARS-CoV-2. However, birds are of special concern as a source of more virulent zoonoses, even though bats harbor the most virulent ones [3]. The World Health Organization describes a zoonotic disease as "A zoonosis is any disease or infection that is naturally transmissible from vertebrate animals to humans". Thus, non-humans play a vital role in the foundation and propagation of zoonotic pathogens. It is estimated that 61% of pathogens infect humans, and most emerging infectious diseases are zoonotic [4,5]. This trend seems likely to continue, as diminishing biodiversity reduces the natural control on the abundance of wildlife pathogens [6], bushmeat consumption continues, and urbanization and the proliferation of agriculture into formerly undeveloped ecosystems bring wild and domesticated animals into greater contact, thus facilitating

spillover of pathogens [7–9]. Factory farming deserves special attention as it is characterized by intensive production systems, close contact between humans and animals, and high population density in monoculture [10]. Nevertheless, spillovers are rare [11] but non-negligible. Yet, most zoonotic diseases begin in wildlife [12], then typically jump to domestic animals and from there to people [13].

## 2. The Distinctions between Natural and Unnatural Systems

In natural ecosystems, biodiversity tends to reduce the incidence of established pathogens by the action of predators, natural enemies, or competitors of the pathogens and their reservoirs and vectors. The abundance of different species, including predators, limits individuals' abundance and population density within single prey species [6], decreasing the opportunity for pathogen transmissions. What transmission occurs tends to be between phylogenetically similar species as pathogens [14–16]. With the reduction of transmissions, there is consequently a reduction in the opportunity for mutations. What mutation occurs mainly consists of increased transmissibility and lower virulence within that given host species. Milder illness supports host mobility and lengthens host survival, increasing opportunities for transmission to other hosts [17]. Indeed, the Omicron variant of SARS-CoV-2 fits this pattern because it emerged after two years of furious transmission and mutation. It is the mildest [18] and most contagious [19] variant identified to date. This tendency of pathogen evolution to become milder is contingent on the pathogen's advantage to spread if the host can walk to the next susceptible host. Mutations that give rise to milder pathogens will be more likely to spread [17].

In disrupted ecosystems (unnatural systems), the lost balance between predators and prey causes an abundance of individuals within certain species, leading to a greater likelihood of pathogen transmission [7]. Thanks to mutations and farmland encroachment, this transmission can potentially jump to domesticated animals that intrude within the proximity of such disrupted ecosystems [7–9]. Although transmission happen more easily to more closely related host species [14,15], these could eventually occur to species more distant phylogenetically. For example, strains of influenza A viruses (IAV) naturally harbored in the wild avian waterfowl replicate poorly in most mammalian hosts [20]. But mutations and opportunities of contact can improve adaptation to mammalians.

Industrialized animal farming systems congregate one or few species in a small space, increasing the opportunity for transmission and mutations and, therefore, the potential generation of new pathogens via antigenic drift. These pathogens could further change via antigenic shift by exchanging genetic material with other viruses (reassortment), sometimes from different host species, in ways that facilitate transmission, enabling a proliferation of mutations potentially harmful to animal health and humans [10,21,22]. For instance, the 2009 Swine Flu Pandemic virus (H1N1) had multiple mutations from avian strains [20].

The industrialization of animal farming represents the antithesis of the diversity and low populational density of natural ecosystems, which is increasingly scarce today. While the origin of spillover of the pathogens that cause HIV, Ebola, and Nipah virus reveals that spillovers can sometimes happen directly from the wild to humans, such cases, however, are less common and, when occurring, can be explained in large part by the disruption of balance in natural ecosystems [8,9]. Although well-preserved natural ecosystems do not guarantee protection from outbreaks for humans [21], systems with an abundance of individuals of a few species with high stocking density increase risk [10,21].

## 3. Worldwide Demand for Animal Protein

The world population and corresponding worldwide demand for animal protein will increase in the coming decades [23]. Industrial farming offers an attractive potential due to its high productivity. However, the attending growth of industrial animal farming and urbanization of the human population in the face of climate change, globalization, and deforestation, which increases the geographical range of vectors, transport of vectors and pathogens, and loss of biodiversity, respectively, are congealing in a perfect storm that

may increase the frequency of outbreaks. Assuming the animal farming trend of confined mono-species operations is unlikely to cease, mitigating its potential effects on pathogen transmission is urgently needed. Biosecurity standards, risk assessment, and monitoring in animal feeding lots are already in place in many industrial farming settings. And when diseases are detected, animals are culled to stop diseases from spreading. These practices and measures should be implemented, and even improved, in the totality of farms using industrial production models, including developing countries.

It is imperative to closely monitor those species traditionally more prone to host potential zoonotic pathogens both in the wild and in domestic settings through all operation phases (i.e., breeding, growing, and even transportation), especially domestic species whose production style involves higher stocking density of homogeneous populations (mono-species) [10,21]. Such characteristics are found in poultry and swine. Another common feature is that both species can harbor IAVs. A species of bird, the wild avian waterfowl, is the natural reservoir, and the pig is the mixing vessel for different strains. Because IAVs have ribonucleic acid as their genetic material (RNA viruses), they have higher mutation rates than DNA viruses. Therefore they can more easily adjust to human infection and transmission [24]. Pork and chicken, compared to beef, deserve perhaps more attention in the face of a worldwide animal protein demand for growth due to their lower cost. Furthermore, compared to pork, a bigger proportion of beef worldwide originated from low-density population systems such as pasture-based farms, which offers less opportunity for pathogen transmission. Animal production based on sustainable pasture-raised models is awakening as poultry and swine producers become aware of these systems' environmental benefits, lower initial costs, and marketing opportunities.

## 4. Virulence and Contagiousness

Pathogens that have evolved to be mild and contagious in one species may be severe or lethal but less transmissible in a new species host [25]. However, the genetic similarity between different species may mean that some pathogens are well-prepared to perform well in a new species [14]. For example, HIV was immediately highly transmissible when it jumped from chimpanzees to humans [26]. Mammals, more than other classes of animals, have passed zoonotic diseases [2]. Considering how confined feeding operations lessen the tendency of pathogens to evolve toward mildness through stocking, pigs, cows, and other mammals raised for meat represent a significant threat. Indeed, swine farming may pose the greatest threat, as pigs are much like humans, mammals, monogastric, and omnivores [27], which may translate to greater overlaps in pathogenic vulnerability [22]. Influenza A viruses are particularly interesting as many can infect swine. Swine have been identified as a mixing vessel for avian, swine, and human influenza strains [28–30]. Swine population's ability to act as a host for many influenza strains from multiple species allows for genetic reassortment and propagation of potentially novel influenza with pandemic potential. This is what occurred with the 2009 H1N1 pandemic.

In addition to livestock's tendency to serve as mixing vessels for pathogens, confined animal operations are often close to areas with dense populations of humans. Modern cities, characterized by a high mono-species populational density, could further favor pathogenic mutations that are highly contagious and adapted to human bodies. In an increasingly globalized economy, zoonoses can readily spread from one crowded urban center to many [31]. This may suggest a need for cities to encourage telework and outdoor activities and improve ventilation in public buildings and on public transportation.

Influenza viruses are endemic in commercial swine populations worldwide. They are also endemic in smaller-scale backyard operations. The latter, while perhaps less conducive to pathogen transmission, lacks the monitoring and biosecurity measures of the former. The spread of influenza viruses from swine-to-swine farm workers with whom they have intense and frequent contact probably occurs frequently. Indeed, the increased antibodies to swine-origin influenza reported for swine workers or those with close pig exposure suggest they are at increased risk of zoonotic transmission [32–34].

Further, the transmission between swine and attendees of agricultural fairs can serve as a spillover point. However, hypothetical spread rates have yet to be quantified and are likely to vary depending on many factors. Fortunately, transmission from pigs to farm workers does not appear to lead to continuous chains of transmission between humans. However, this also has yet to be precisely quantified.

The most recent swine-related influenza pandemic occurred in 2009. This is a concern considering that the records show that during at least the last three hundred years, influenza pandemics have occurred roughly every 20 to 30 years [35]. Since the 20th Century, at least five influenza pandemics for which swine played a critical role have occurred [36]. There are estimates of at least 50 million deaths caused by the Spanish flu of 1918 [37]. The world population at this time was about 1.7 billion. The recent Swine Flu of 2009, a triple-reassortant with genes from avian-swine-human, is estimated to have infected about 60.8 million people in the United States [38] and about one to three billion people worldwide, which is about 15–45% of the world's population [39].

While swine are considered the mixing vessel of flu viruses that cause pandemics, sequence data suggests that the 1918 pandemic virus was likely derived from a wild waterfowl IAV [37]. It might have been an H1N1 avian precursor for an H1N1 swine virus that might have caused the 1918 pandemic [40]. Indeed, in 1930, the first time an influenza virus was isolated was from swine. It was an H1N1 subtype from the same lineage as the 1918 pandemic virus [41]. According to Taubenberger and Morens [37], the 1918 virus, later known to be an H1N1 IAV, is believed to have reassorted with an avian IAV again to cause the pandemics of 1957 (H2N2), 1968 (H3N2), and 2009 (H1N1) [37]. Thus, wild birds and domestic poultry deserve much attention, along with swine. An H1N1 virus caused a pandemic of an unknown source that occurred in 1977. The seasonal human influenza virus (H1N1) is also a descendant but through antigenic drift of the 1918 influenza virus (H1N1). H1N1 and H3N2 influenza viruses are currently the major causes of seasonal influenza [42].

The H1N1 lineage was the sole cause of influenza in swine in the USA since 1930, when it was first isolated, until 1998, when an antigenic shift resulted from reassortment with avian and human influenza to generate an updated version of H3N2 [43]. Variants of H3N2 and the classical H1N1, along with a reassortant between them, the H1N2 virus, have been the cause of influenza in the North American swine population since 1998 [41]. Other cases of reassortment of IAVs in swine have been detected, but these have not been isolated on farms other than where they were initially found [41]. The different subtypes of IAV that currently circulate in the swine population are commonly differentiated according to their different surface glycoproteins, but they all contain similar genetic compositions [41]. If these viruses continue to spread, they will increase the opportunity for antigenic shifts among the swine population. With this, new reassortant viruses can emerge that could threaten the world's human population.

Regarding avian influenza, the first description dates back to 1878 in northern Italy; however, it was not until 1955 that it was identified as an IAV [44]. Several subtypes of highly pathogenic avian influenza have been detected in recent years worldwide, and different species of wild and domestic birds and mammals [45]. Since 2003, sporadic human cases of the H5N1 subtype have been reported worldwide [46] and since 2021 in the USA [47]. These human cases reported in the USA followed exposure to infected poultry. And as of 11 March 2023, one human case, 6356 wild birds, and 58,602,281 poultry of a new globally circulating H5N1 subtype have been reported in the USA [48]. No human-to-human transmission has been demonstrated with the current circulating H5N1 subtype. However, there is an investigation trying to confirm the source of the infections of a child, who died, and her father, in Cambodia with a different H5N1 subtype [48]. Although experts believe these cases resulted from direct exposure to infected birds [49], there is the fear of reassortment with human or swine viruses. Such an event could increase the possibility of human-to-human transmission and the risk of a new pandemic. Research on these new viruses in the swine and bird population can offer critical information to

understand the transmission principles between different species [43], thereby preventing the emergence of new viruses with pandemic potential.

Antigenic drift, on the other hand, might be more prone to generate milder but more infectious viruses, as hypothesized by evolutionary ecologists such as Paul Ewald [17]. As stated before, according to this hypothesis, under natural conditions, pathogens tend to evolve toward becoming milder because their transmission opportunity increases with the mobility of hosts. Therefore, it is also imperative to research antigenic drift, as it may determine the spreading, thus the overlap of different viruses, which allows for an increased chance of reassortment.

As mentioned before, the concern about generating new pathogens with pandemic potential is even more problematic considering that pandemic-favorable conditions such as deforestation, industrial farming, urbanization, climate change, and globalization are rising. Based on history, this article suggests that the likelihood of a swine flu pandemic is not negligible. Therefore, despite the availability of technical solutions to the problem, we need targeted and cost-effective surveillance, mitigation, and control measures, both in industrial farms and in cities (and even in natural ecosystems nearby domestic animal and human settings), to reduce the impacts of the emergence of influenza viruses with pandemic potential.

Controlling the chance for pathogen emergence requires a better understanding of how animal farming technologies and human-animal-ecosystem interactions create opportunities for evolution and spillovers and how fast and often those occur. Identifying these interactions at the landscape level can greatly benefit from remote sensing technology and geographical information systems, further helping holistic management for mitigation and prevention [50]. Such knowledge not yet existing is required to stop a pandemic at its very origin or, in the worst case, control and mitigate its distribution and consequences. In line with the adage, an ounce of prevention is worth a pound of cure, the most practical approach would be to prevent spillover opportunities, both from wildlife to domestic animals and from animals to humans, but all mitigation approaches are welcomed.

## 5. Conclusions

As vaccines and treatments for COVID-19 have emerged [51,52], the world has become accustomed to the COVID-19 pandemic [53]. However, the world needs to better prepare for the next pandemic, and many factors are concerting to speed up the day of such an event. Urbanization and its growing global population demand an ever-increasing supply of animal protein, which is being progressively produced by intensive animal operations characterized by processes with a high populational density of a single species of confined animals. Fortunately, current confined animal operations usually count on suitable technologies for monitoring and biosecurity. Although biosecurity measures are better controlled in commercial poultry and swine operations than in backyard or free-range farming, the latter might pose less risk regarding its lower total populations and stocking densities. Concern still needs to be addressed about potential failures in the routine biosecurity protocols or procedures in industrial farming systems of developing countries with less technology. Not counting the ecosystem control measures provided by natural biodiversity, models of the high populational density of a single species of animals, especially pigs and poultry, generally increase the opportunity to transmit pathogens. In turn, more opportunity for transmission translates into more pathogen evolution and reassortment opportunities that could result in new strains.

There is a concern about the possibility of generating highly infectious pathogens, virulent and well-suited to infect and spread among humans. As more humans live in urban settings, also characterized by a high density of a single species (humans), the conditions continue for an increased chance of pathogen evolution. The more the opportunity for transmission and availability of susceptible hosts due to the high population density, the more possible the evolution towards greater infectiousness and even virulence. This principle should be considered; in the face of other spillover risk factors that increase

the likelihood of devastation, such as climate change, globalization, deforestation, and increased travel and modern transportation that paves the way for pathogen spread. New technologies and guidelines for animal agriculture should complement changes in urbanization, ventilation, working environments, transportation, and public space.

**Author Contributions:** Conceptualization, M.J.M.-M.; literature collection and curation, M.J.M.-M. and E.K.; writing—original draft, M.J.M.-M.; and writing—review and editing, M.J.M.-M. and E.K. All authors have read and agreed to the published version of the manuscript.

**Funding:** The Hampton and Esther Boswell Distinguished University Professorship.

**Institutional Review Board Statement:** Not applicable.

**Informed Consent Statement:** Not applicable.

**Data Availability Statement:** Not applicable.

**Acknowledgments:** We thank Tom and Cheryl Boswell for funding the Hampton and Esther Boswell Distinguished University Professorship position, which made this work possible. The findings and conclusions in this manuscript are those of the authors and do not necessarily represent the views of the Hampton and Esther Boswell Distinguished University Professorship or DePauw University.

**Conflicts of Interest:** The authors declare no conflict of interest.

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
