# Peer review of "Stocking Density and Homogeneity, Considerations on Pandemic Potential"

_zoonoticdis, doi:10.3390/zoonoticdis3020008_

Round 1

Reviewer 1 Report

The manuscript is well written and the topic actually. 

Anyway some changes have to be permormed before  going ahead in the pubblication phase. If authors well follow the suggestion given below I will certainly recommend this case report for pubblication on the Journal.

First, the authors should consider in the introduction the impact of the contact human-wildlife with the spreading of differrent types of diseases. Not only from wildlife to humans but aslo from wildlife to domestic animals. This aspect in very important in foodborne diseases, in particular.  I suggest this because One Health is very important nowadays.

I advice you also to consider the use of GIS in study of zoonoses.

Here the following articles that I strongly advice you to include: 

https://doi.org/10.1007/s10344-015-0974-9

https://doi.org/10.1111/tbed.13916

 https://doi.org/10.3390/app13010390

- https://doi.org/10.3390/rs12213542

- https://doi.org/10.3390/cli9030047

DOI: 10.20506/rst.32.2.2242 

Moreover, I suggest to improve the conclusion with the arguments presented and give also more suggestions for a correct management of the problem in the future (for example reporting pratical news technologies that should complement change in ventilation, urbanization and public space). 

Bibliography must be improved with the mentioned above references. 

Author Response

We strongly appreciate the corrections made. We have addressed them. Please see in the attached word document our explanation of how we addressed each correction. Likewise, please see the corrected manuscript draft, where the changes are tracked to identify the modifications made according to your suggestions easily. 
We are sure these corrections have substantially improved our manuscript. 
Thanks so much. 

Reviewer 2 Report

The paper provides insight into important issues related to the growing trend of outbreaks of zoonotic diseases worldwide. The paper discusses the disease outbreaks in light of the interaction of the host, environment, and disease agent. The risk of mutation and the generation of new pathogens, which are important issues of the disease outbreak in the actual global situation, are discussed and illustrated. There are given epidemiological determinants of the disease outbreaks such as industrial farming, diminishing biodiversity, bushmeat consumption, urbanization, the proliferation of agriculture, disrupted ecosystems, loss of biodiversity, host mobility, natural ecosystems, the abundance of individuals of few species, crowded urban centers, etc. The paper is interesting for the international audience as it gives recommendations for scientific research focus and drives to the important issues of eventual disease outbreaks that could be faced in the future.

Author Response

Thanks so much for reviewing our manuscript. 

Reviewer 3 Report

This paper discusses a general view of population density and biodiversity as it relates to disease transmission and pandemic potential. It asserts that biodiversity, urbanization, high population density of a single species, and industrial farming favors pathogen transmission, evolution and increased virulence. However, no concrete data to support claims were cited or presented.

Line 31- a more recent infectious disease outbreak disease data is needed (not 1980-2013), especially considering the most recent Covid-19 pandemic

Line 61 - can you give an example for this claim?

Line 62-64 - what about wild waterfowl which is considered the natural reservoir of Influenza A viruses, and a primary/continuous source of virus for other animal species including humans?

Line 65 - data to support claim?

Line 71-73 - most big poultry operations don't have much human contact.

Line 79-80 - when a disease outbreak occurs, chickens/turkeys are culled to stop disease spread.

Line 118-119 - while the pig is considered the mixing vessel of flu viruses that cause pandemics, sequence data suggests that the 1918 pandemic virus was likely derived from a wild waterfowl influenza A virus (Taubenburger & Morens 2020 - your #31 reference). So, maybe wild birds and domestic poultry should be given attention as well?

Line139-140 - Spanish flu pandemic happened in 1918, not 2018!!

Line 141 - Swine flu of 2009 was actually a triple-reassortant with genes from avian-swine-human.

Line 145 - Covid19 is the most recent pandemic, not 2009 swine flu

Line 165-168 - these 2 sentences can be put together as 1!

Line 169-176 - biosecurity measures are better controlled in commercial poultry operations than in backyard or free-range farming.

Line 177-187 - also consider increased travels and modern transportation that paves the way for pathogen spread.

References are quite outdated. Reference #15 and #26 are duplicates.

Author Response

(The authors gave the same response as above.)

Reviewer 4 Report

The manuscript briefly overviews the zoonotic agents and their pandemic potential in a population of high density. A few examples of influenza outbreaks are given in the manuscript, proving the potential threat posed by swine to humans. The Pandemic has recently become a very popular topic, it is crucial to put more stress on different zoonotic diseases other than Covid 19 and their potentially hazardous effects on humans.

I have just two comments for the authors.

1.      I suggest giving some more details about the influenza outbreak among swine, there was no information about the evolution of swine H3N2 Influenza in the USA in 1998.

2.      The statement “This is what occurred with the 2009 H1N1 pandemic’’  is too short, I encourage the authors  to explain what happened with the H1N1 virus in 2009 

Author Response

(The authors gave the same response as above.)

Round 2

Reviewer 3 Report

*2 new inserted paragraphs (lines 61-79) are very confusing and since it comes from only 1 reference (#5), maybe can be summarized? Sentence on lines 62-66 is too long and vague.

*Are you sure about the statement in lines 128-129? And by the way, I don't think the 1918 pandemic is "now known as swine flu because it was first isolated from swine in 1930" (line 183). Maybe if there was better technology before 1930, it would have been called "bird flu", since sequence data suggest that the flu virus was likely derived from wild waterfowl? And close scrutiny of paragraphs in lines182-196, somehow points out to "1918 H1N1 IAV" as the "mother virus" from which reassortants came from to cause the pandemics of 1957, 1968, 1977,2009, etc? So, maybe it is prudent to look more closely on the very origin of these pandemics, which are the wild birds (natural reservoir of Influenza A) and passed on to domestic poultry, to swine, then to humans? High path AI is actually posing a big threat to wild birds and domestic poultry globally right now. One human infection with AI in Colorado; 2 (1 died Feb 22, 2023) in Cambodia.

*References are bungled (#19, #20); #25 is empty; #29 & 40 are duplicates

Author Response

We strongly appreciate your corrections and suggestions. We believe they greatly improved the quality of the manuscript. We agree with all your comments, and we corrected point by point the manuscript accordingly. Please attached word document.
